# Mesothelioma Malignancy and the Microenvironment: Molecular Mechanisms

**DOI:** 10.3390/cancers13225664

**Published:** 2021-11-12

**Authors:** Francesca Cersosimo, Marcella Barbarino, Silvia Lonardi, William Vermi, Antonio Giordano, Cristiana Bellan, Emanuele Giurisato

**Affiliations:** 1Department of Biotechnology Chemistry and Pharmacy, University of Siena, 53100 Siena, Italy; francesca.cersosi@student.unisi.it; 2Department of Medical Biotechnologies, University of Siena, 53100 Siena, Italy; marcella.barbarino@unisi.it (M.B.); antonio.giordano@unisi.it (A.G.); cristiana.bellan@unisi.it (C.B.); 3Sbarro Institute for Cancer Research and Molecular Medicine, Center for Biotechnology, College of Science and Technology, Temple University, Philadelphia, PA 19122, USA; 4Department of Molecular and Translational Medicine, University of Brescia, 25100 Brescia, Italy; silvia.lona@gmail.com (S.L.); william.vermi@unibs.it (W.V.); 5Department of Pathology and Immunology, Washington University School of Medicine, St. Louis, MO 63130, USA; 6Division of Cancer Sciences, School of Medical Sciences, Faculty of Biology, Medicine and Health, The University of Manchester, Manchester M13 9PL, UK

**Keywords:** mesothelioma, tumor microenvironment, inflammation, macrophages, cancer stem cells

## Abstract

**Simple Summary:**

In the tumor microenvironment, interaction among tumor cells, immune cells, stromal cells, and the extracellular matrix is vital to support pro-tumor mechanisms such as drug resistance and metastases. Malignant pleural mesothelioma has a unique and complex tumor microenvironment. Several reports underlined the key role of immune and stromal cells in tumorigenesis and progression of mesothelioma. These non-cancer cells, via a reciprocal informational exchange with tumor cells, established a chronic inflammatory microenvironment that support the malignancy and the chemoresistant phenotype of the tumor. The knowledge of the cellular and molecular mechanisms underlying tumor microenvironment interconnection was recently considered a crucial point for the design of more effective therapeutic strategies. In this review, we summarize the molecular mechanisms by which stroma and immune cells support the malignancy of mesothelioma and their potential therapeutic targeting.

**Abstract:**

Several studies have reported that cellular and soluble components of the tumor microenvironment (TME) play a key role in cancer-initiation and progression. Considering the relevance and the complexity of TME in cancer biology, recent research has focused on the investigation of the TME content, in terms of players and informational exchange. Understanding the crosstalk between tumor and non-tumor cells is crucial to design more beneficial anti-cancer therapeutic strategies. Malignant pleural mesothelioma (MPM) is a complex and heterogenous tumor mainly caused by asbestos exposure with few treatment options and low life expectancy after standard therapy. MPM leukocyte infiltration is rich in macrophages. Given the failure of macrophages to eliminate asbestos fibers, these immune cells accumulate in pleural cavity leading to the establishment of a unique inflammatory environment and to the malignant transformation of mesothelial cells. In this inflammatory landscape, stromal and immune cells play a driven role to support tumor development and progression via a bidirectional communication with tumor cells. Characterization of the MPM microenvironment (MPM-ME) may be useful to understand the complexity of mesothelioma biology, such as to identify new molecular druggable targets, with the aim to improve the outcome of the disease. In this review, we summarize the known evidence about the MPM-ME network, including its prognostic and therapeutic relevance.

## 1. Introduction

Malignant pleural mesothelioma is the main cancer affecting the pleural membranes covering the lungs. It is considered a disease of the elderly and generally manifests in an advanced stage after decades from environmental carcinogen exposure [1]. In addition, its intrinsic heterogeneity, lack of effective targeted therapies and still insufficient knowledge of MPM biology, compromise the quality of life and the prognosis of MPM patients. For sixteen years the only approved therapy has been the combination of platinum and antifolates; recently, nivolumab, a PD-1 blocking antibody, in combination with ipilimumab, a CTLA-4 inhibitor, has also been approved as first-line therapy for unresectable MPM [2]. MPM pathogenesis has unique features because it is mainly related to the exposure of external carcinogens that are mostly represented by asbestos fibers [3,4]. Despite the efforts to limit asbestos use, it is actually banned in 30% of countries. For this reason and for the increasing concerns about the carcinogenicity of new fibrous materials similar to asbestos, the incidence of mesothelioma is not expected to decrease in the coming years [5]. In addition, cases of mesothelioma have also been reported in chronic inflammatory conditions, in absence of fibers exposure, such as chronic pleural diseases, chronic empyema or therapeutic pneumothorax [6].

MPM is characterized by a low tumor mutation burden (TMB) [7], uncommon genetic aberrations, and recurrent somatic mutations in tumor suppressor genes, in both asbestos and non-asbestos induced tumors [8]. The first and most common mutation described in mesothelioma is the deletion of the Cyclin D dependent Kinase inhibitor 2A (*CDKN2A*) gene on chromosome 9 [9], accounting for approximately 70% of MPM cases [10]. The deletion of this gene affected the cell cycle regulating function of pRB and p53. For its proximity to *CDKN2A*, the methylthioadenosine phosphorylase (*MTAP*) gene is frequently co-deleted in different cancer types, including malignant mesothelioma [11,12]. Other common mutations in mesothelioma are in chromosome 3, involving the loss of the *BAP1* gene, in chromosome 22 enclosing the neurofibromin2 (*NF2*) gene, and in *TP53* [7,13,14,15]. BRCA1-associated protein–1 (BAP1) has many biological activities, including genome stability, DNA damage repair, modulation of the cellular metabolism, regulation of transcription and cell death, among others. *BAP*-*1* loss, together with *MTAP*/*CDKN2A* deletion, has been recently proposed as useful markers to improve the diagnostic sensitivity for MPM [16] (Figure 1). Mutation in the *NF2* gene, encoding the cell growth-regulating protein Merlin, has been described in about 50% of MPM [17], and has been linked to mesothelioma progression. Alteration in NF2 function has been recently related to the tumor immune microenvironment and proposed as biomarker for MPM patient’s stratification for immune-checkpoint blockade (ICB) therapies [18]. It has been reported that in BNC mice, where the specific disruption of the *Bap1*, *Nf2*, *Cdkn2ab* tumor suppressor loci in the mesothelial lining of the thoracic cavity leads to a highly aggressive MM, an infiltration of leukocytes was found [19]. In particular, a significant number of macrophages, T cells, including regulatory T cells (Tregs), B cells and NK cells was observed. This recapitulates the histological features and gene profile observed in human patients carrying combined *BAP1*, *NF2* and *CDKN2A* alterations [20], indicating that the combined deletion of these tumor-suppressor genes creates a mesothelioma-specific microenvironment. The enrichment of NF-kB signaling pathway in BNC tumors likely contributes to the recruitment of immune cells to these tumors. In support of the link between TME and different genetic background, Yang H and collaborators provided evidence that CD8^+^ T cells were mainly enriched in the MPM harboring Large Tumor Suppressor Kinase ½ (*LATS1*/*2* mutation compared with *NF2*-mutant cancer. In addition, MPM tumors harboring *LATS1*/*2* mutation is associated with high PD-L1 expression and rather than *NF2*-mutant MPM, display enriched Tregs signature and plasma B cell signature [18], suggesting that different tumor-infiltrating immune cell patterns exist between dysregulation of NF2 and Hippo-YAP signaling in MPM. More recently, the relationship between p14/ARF encoded by *CDKN2A* and tumor microenvironment was evaluated. Pezzuto et al. found that p14/ARF-negative tumors are characterized by a high percentage of CD163^+^ cells and low PD-L1 and CD4 expression, correlated with an immune microenvironment less sensitive to immune checkpoint inhibitors [21]. Collectively, these data support the idea that genomic intratumor heterogeneity shapes the MPM tumor microenvironment and modulates host immune surveillance or immune escape in MPM [22].

According to the low TMB, microsatellite instability and deficiency of DNA mismatch repair system proteins (dMMR) (MutS Homolog 2 (MSH2), MutS Homolog 6 (MSH6), MutL Homolog 1 (MLH1)) due to gene mutation or epigenetic silencing, have been found in a small subset of MPM [23,24,25]. In this scenario of a low mutational burden, epigenetic regulation is considered to significantly contribute to the malignant mesothelial transformation. Particularly for cancers linked to environmental insults such as mesothelioma, the study of epigenetics has been demonstrated to be a useful indicator of disease risk, with diagnostic and prognostic value [26,27,28,29]. However, the deregulation of genes and protein expression described above contribute to, rather than determine, the MPM onset. Different from other cancers, driver mutations have not been found in mesothelioma and the pathogenesis of this cancer is undoubtedly related principally to the inflammatory microenvironment created by asbestos deposition in the pleura that stimulates the immune response. A crucial role in this process is recognized to the pleural macrophages recruited at the inflammatory site, which, failing the attempts to eliminate the fibers, are subject to frustrated phagocytosis, a process leading to activation of Nicotinamide adenine dinucleotide phosphate (NADPH), the generation of reactive oxygen species (ROS) and release of proinflammatory molecules (IL-1β, IL-8, IL-6 and TNF-α) [30]. Accumulation of asbestos fibers leads to aberrant activation of intracellular pathways and transcriptional processes responsible for the malignant transformation of mesothelial cells and the development of a unique inflammatory microenvironment [31]. Indeed, tumor-associated macrophages (TAMs), T regulatory cells (T_reg_), such as cancer-associated fibroblasts (CAFs), are the most abundant population of MPM infiltration, that, in response to pro-tumoral signals, acquire malignant and immunosuppressive properties, influencing the progression of the tumor [32] (Figure 1). Tumor cells have developed different mechanisms to escape immune surveillance, such as the activation of inhibitory pathway (PD-1/PD-L1) leading to T-cell exhaustion and suppression of cytotoxicity. Overexpression of PD-L1 on immune cells and PD-1 of tumoral-origin was also described in MPM [33]. Moreover, resistance to anti-cancer drugs is another feature of MPM cells, that is supported by the TME crosstalk [34]. Given the importance of the interactions between the surrounding and neoplastic cells, several research groups have investigated the content of TME in terms of cellular component and exchanged soluble factors with the aim to improve the immunotherapy. However, its variability contributes to the complexity and the difficulty to evaluate it in terms of diagnosis, prognosis and therapeutic approach [32]. According to Linton et al., inflammation in mesothelioma could be considered a “friend” or a “foe”, because chronic inflammation could be established as a favorable environment for cell survival on one hand and could induce the suppression of anti-cancer response on the other hand [35]. For this, the inflammatory landscape of mesothelioma TME is required to be deeply investigated in order to restore the immune response and develop strategies with better therapeutic benefits. In this review, we summarize the known evidence on the MPM TME, focusing on the inflammatory population, the interplay between immunity, stroma and tumor cells, such as mechanisms regulating immune-evasion and drug resistance.

Additionally, several studies reported the prognostic value of immune cells in the mesothelioma microenvironment. High incidence of CD8^+^ T cells was associated with a better prognosis and favorable outcome; on the contrary, the presence of M2-like macrophages that represent the most abundant immune population in MPM-TME, was correlated with a worse prognosis and no improvement in the overall survival [36,37]. Thus, here we described the potential prognostic role of these cells and highlight the importance to understand the MPM-TME players as potential candidates for promising mesothelioma therapy.

## 2. Mesothelioma Stem Cells (MSCs) and Chemoresistance

A sub-population of cancer cells, indicated as cancer stem cells (CSCs), was assumed to be involved in the tumorigenesis and metastatic progression. These cells display the capability of self-renewal via asymmetrical division, maintaining the stem niche in tumor sites. CSCs theory is based on a hierarchical organization of cells in the tumor, where only a sub-set of cells with high tumor-forming properties can give rise to a heterogenous cancerous cell population associated with rapid recurrence [38,39]. Different studies identified the presence of CSCs within malignant mesothelioma (MM) tissue as responsible for tumor heterogeneity [40], chemoresistance and relapse after therapy [41]. MM, as with other tumor types, is strongly characterized by inter and intra-tumor heterogeneity and the persistence of resistant CSCs in tumor tissue represent the main cause of treatment failure in cancer therapy [42,43]. Kay K et al. demonstrated that cisplatin-based treatment upon MM cell lines favored the selection of a side population with increased replicative potential and high expression levels of stemness-related genes, including *OCT4*, *NOTCH1* and *BMI1* [44]. The presence of *SOX2*/*OCT4* positive cells in chemotherapy-resistant MM has been later confirmed by Blum et al. who additionally demonstrated the tumorigenic properties of these cells by in vivo studies [45]. Various tumors have shown the presence of a chemoresistant sub-population of cells expressing high levels of ABC-transporters protein and increased aldehyde dehydrogenase (ALDH1) activity [46]. Similarly, overexpression of ABCGE protein was found in MM cells expressing stem-like characteristics, such as a therapy resistant phenotype [44] and an elevated ALDH1 activity characterized by cells in MM with CSCs properties [47]. Up-regulation of ABCB-5 drug transporter was observed in MM CSCs as necessary for the acquisition and maintenance of a stemness and chemoresistant phenotype. Additionally, intrinsic mechanisms, such as increased expression of the autocrine loops Wnt/GSK3β/β-catenin/c-myc in MM CSCs support the over-activation of ABCB-5 as well as drug resistance of these cells [48]. Conino et al. have demonstrated that Pemetrexed induced rapid senescence in MM cells associated with the production of cytokines and pro-inflammatory molecules as well as the involvement of STAT3 signaling, that, in turn, activate epithelial-to-mesenchymal transition (EMT) programs, via the release of invasion-promoting factors (MMP-2) and the emergence of chemoresistant, clonogenic and ALDH^+^ cells [49]. The deficiency of the tumor suppressor gene, *NEF2*, in mesothelioma mouse models was correlated with an increase of CSCs in the tumor tissue [50]. Recent studies demonstrated a link between the presence of *NF2*-negative cell populations and high sensitivity to FAK-signaling inhibitors in MM cells [51]. The tumorigenic role of FAK and its contribution to self-renewal and aggressiveness of CSCs in tumors has been widely reported [52]. CSCs elimination by using FAK inhibitors treatment in MM cell lines and NOD/SCID mice injected with MM stem-like cells has been observed. Since CSCs constitute the main targets for drug resistance, a combined approach with FAK inhibitors and chemotherapeutic agents could represent a new potential cancer treatment strategy to eradicate CSCs and overcome the mechanism of chemoresistance [45,51]. Nowadays, less is known about molecular mechanisms underlying CSCs chemoresistance in MM. Therefore, a better characterization of MM CSCs biology is required to fight drug resistance and improve MM treatment.

## 3. Mesothelioma Stem Cells Malignancy and TME

Different studies have highlighted the metastatic ability of CSCs as key players of tumor progression [53], also in MM [54,55,56]. In this regard, biochemical analysis has shown that spheroid-derived mesothelioma stem-cells (MSCs), expressing increased levels of the cancer cell survival-related protein transaglutaminase (TG2), had more invasive and migratory capabilities compared with monolayer-derived mesothelioma cells. TG2 has been reported to have important roles in CSC-phenotype acquisition and invasiveness of tumor cells. Inactivation of TG2 has been shown to decrease the expression of epithelial-to-mesenchymal transition markers (Fibronectin, MMP-9, Slug and Snail), the Matrigel invasive abilities and increased the levels of the pro-apoptotic factors as caspase-9 and PARP activity [55]. Moreover, a recent study demonstrated that inhibition of YAP1/TAZ/TEAD signaling pathway in MSCs from peritoneal and pleural-derived mesothelioma cells reduced migration and invasiveness of these cells, such as increased pro-apoptotic markers expression and negatively affected the spheroid forming ability of these MM stem-like cells [56]. A heterogenous population of cells, expressing stromal, immune e stem-cell markers, was identified by the analysis of the tumor spheroid derived from orthotopic MM murine model [54]. Several reports have documented the importance of TME interactions to sustain tumor growth and metastasis [57]. Indeed, a study demonstrated that stem cells number in tumor spheroid raised in response to chemotactic signals, specifically increased expression of the SDF1/CXCR4 axis, which induced the recruitment of these cells at the tumor sites [54]. These data highlight the potential impact of investigating TME signaling in order to find targetable molecules involved in cancer cell dissemination to secondary sites.

## 4. Mesothelioma Microenvironment Crosstalk: Molecular Mechanisms

### 4.1. Mesothelioma and Stroma

Tumor growth depends not only on cancer cells activity, but also by the interactions between neoplastic cells, stroma, extracellular soluble factors, and inflammatory cells that collaborate to support the tumor progression. Tumor cells are able to change the surrounding microenvironment by the release of soluble factors that induce the malignant transformation of resident stromal cells. The communication with the stroma is necessary to create a permissive environment allowing cancer cells to escape immune defensive mechanisms and to metastasize [58,59]. Thus, understanding this complex network that orchestrates the TME is required for the discovery of promising therapeutic target therapy. Different studies have investigated the content of MPM tumoral stroma, as a source of prognostic markers and potential therapeutic targets. In mesothelioma, as with other tumor types, the stromal component plays an important role to support tumor growth and invasion [60] (Figure 2). Lievense et al. identified high expression of pro-inflammatory soluble cytokines in pleural effusion of MPM patients as well as in MPM cell line supernatant. These cytokines, including IL-6, TGF-β, VEGF and IL-12, are known for taking part in the malignancy of the tumor, including the invasive, angiogenetic, and immunosuppressive mechanisms [61]. Proteins of the extracellular matrix (ECM) resulted in being up-regulated, especially in the most aggressive histological subtypes of MM. Integrins, collagen, fibronectin and metalloproteinases have been highly produced by mesothelioma cell lines to make ECM permissive for chemotaxis and invasion [36]. A previous study identified the involvement of TGF-β pathway in the expression of connective-tissue growth factors (CTGFs) in MM tumor cells and surrounding stromal cells [62]. CTGF expression has been found in several cancer types to be correlated with malignant features, such as angiogenesis, invasion, and metastasis [63]. In MPM, the CTGF expression was found to participate in the modulation of the ECM via secretion of matrix-associated proteins in favor of tumor progression [62]. Cancer-associated fibroblasts (CAFs) are one of the main components recruited in the tumoral stroma. These cells interact with tumor cells, and recruit immune and vascular cells at the tumor site through the release of soluble factors, such as cytokines and chemokines [59] (Figure 2). CAFs are able to remodel ECM by the production of several ECM-related proteins (e.g., integrins) that mediate the crosstalk with tumor cells and promote local invasion and metastatic spread [64]. These spindle-like cells are also found in mesothelioma tissue, associated with pro-tumor functions, such as stromal remodeling and tumor invasion, and are correlated with poor prognosis [65]. Ohara et al. proposed a mechanism by which fibroblasts participate to pleural fibrosis in the early mesothelioma phases in response to ROS production by frustrated macrophages. Activated fibroblasts exerted a pro-tumor role by the expression of CTGF and other cytokines [65]. Li Q. et al. suggested positive feedback between MPM cells and CAFs: tumor cells producing growth factors, such as FGF-2 and PDGF-AA, promote the growth and the activity of CAFs that in response secrete cytokine HGFs enhancing the migratory and invasive abilities of tumor cells [66]. Furthermore, MPM is indicated as a tumor with a high tendency to angiogenesis [67] and it is well known that CAFs are pro-angiogenic factors in tumors [68]. Serum and pleural effusion from mesothelioma patients have shown high levels of angiogenic cytokines as VEGF and FGF-2 that are linked with the development of new blood vessels in tumors [67,69,70]. Additionally, a lower infiltration of immune cells has been correlated with high expression of stromal-related and connective-related genes in mesothelioma tissues [33]. MM contains a heterogenous population of immune cells, changing among patients and histological subtypes [32,60]. Because of the prognostic and therapeutical role of immune cells, the investigation of this area has recently attracted the interest of immune oncology research. Ujii et al., with the aim to characterize MPM TME, identified immune markers with prognostic value in tumor and tumoral stroma of epithelioid MPM tissue. Specifically, they found that markers of tumor-infiltrating lymphocytes (CD8, CD20) correlated with a better prognosis. Instead, high density of M2-like TAMs (CD163^+^) and cytokine receptors (IL-7R^+^) expression by tumor cells resulted with decreased patient survival [71]. Several findings have demonstrated that TME of MPM is enriched with a high number of immunosuppressive cells, among these M2-like TAMs and regulatory T cells represent the most abundant immune cells in the MM microenvironment [32,36]. Pleural effusions of MPM patients have shown a strong infiltration of activated cytotoxic and helper T cells; however, most of them are associated with T cell exhaustion markers, such as PD-1^+^, TIM-3, LAG-3, that negatively regulate lymphocyte activity [72]. Additionally, evidence shows that PD-L1 signaling in tumors promoted T helper-1 (Th-1) cells reprogramming in T-regulatory cells [73]. The high levels of PD-L1 in mesothelioma tissue have made it an eligible tumor for immune checkpoint inhibitor-based therapy [33] (Figure 2). Recently, Klampatsa et al. identified, by flow cytometry analysis on MPM patient samples, a proportion of CD8^+^ tumor-infiltrated lymphocytes (TIL) and tissue-resident memory (Trm) cells with hypofunction that was related not to the expression of inhibitor molecules but to the higher degree of Tregs in TME and to the expression of the Eomes transcriptional factor, known as regulator of CD8^+^ cell functions [74]. A link between high density of CD8^+^ immune cells and improved MPM patient outcomes has also been demonstrated [71,75,76], suggesting a potential therapeutic strategy. Natural killer (NK) cells were also present in MPM samples; however, these cells showed an immunosuppressive profile and lower cytotoxic functions [77]. Given the heterogeneity of immune cell content in MM TME and the pro-tumor functions of the stromal content, an understanding of these components is required for better understanding MM pathogenesis and for the development of an efficacious anti-cancer targeted therapy.

### 4.2. Macrophages in Mesothelioma

Macrophages are a heterogenous population of immune cells acting by phagocytosis and destruction of foreign antigens [78]. Failed phagocytosis of asbestos fibers by macrophages represents one of the possible mechanisms promoting neoplastic transformation of mesothelial cells [79]. Phagocytic macrophages, unable to eliminate fibers, release oxidative molecules and pro-inflammatory cytokines that promote a pro-inflammatory environment and activate signaling pathways in tumor cells that help them to survive despite the asbestos-related damage [80]. A critical pro-inflammatory mediator in the mesothelial transformation process has been identified in high-mobility group protein box 1 (HMGB1), a cytokine that, upon asbestos exposure, is released by mesothelial cells recruiting and activating macrophages. Moreover, HMGB1 impairs macrophage phagocytosis and induces the secretion of tumor necrosis factor-*α* (TNF-*α*), protecting the mesothelial cells from death signals and sustaining the chronic inflammatory response [81,82,83]. The binding of HMGB1 to specific receptors on macrophages activates the NLRP3 inflammasome and induces the secretion of IL-1β, IL-18, IL-1α, and HMGB1 itself, establishing a chronic inflammatory loop [84]. By this mechanism, the new mesothelioma cells are able to proliferate, giving rise to a progeny of neoplastic cells [85]. Additionally, MM tissue has been shown to express high levels of the “don’t eat me” signal CD47 that help tumor cells to escape from the immune surveillance systems, including macrophage phagocytosis [86]. Macrophages are plastic cells that, in response to environmental signals, may acquire different phenotypes [87]. The classical or pro-inflammatory (M1) and the alternative or anti-inflammatory (M2) profiles represent the main phenotypes of polarized macrophages [88]. Tumor-associated macrophages (TAMs) are generally infiltrating immune cells that preferentially polarize to M2-like phenotypes [89]. These M2-like TAMs favor a pro-tumor microenvironment via the production of several growth factors and enzymes promoting angiogenesis, immunosuppression and metastases [90]. In mesothelioma microenvironments, TAMs represent the most abundant immune population, and the high prevalence of these cells has been associated with poor prognosis in mesothelioma patients [36] (Figure 2). The expansion of CD206^+^-M2-like macrophages during mesothelioma progression was observed in tumor tissue of an orthotopic model developed to mimic MPM-ME [91]. MPM cells, by producing high levels of the monocyte chemoattractant protein CCL2, induced the recruitment of monocytes at tumor sites [92]. In various cancers, the activation of the CCL2/CCR2 axis, that mediates the crosstalk between TAMs and tumor cells, was associated with metastases and cancer progression. Moreover, targeting CCL2 in lung cancer has been shown to reduce the macrophages recruitment and the M2-polarization rate, such as with active cytotoxic T cells [93]. A previous study described MM cells as able to induce the M2-like polarization, the release of pro-inflammatory cytokines (TNF-α; IL-10) and the acquisition of an immunosuppressive profile [94]. Indeed, high expression of immunomodulatory cytokines. including TGF-β, IL-10 and M-CSF, have been found in pleural effusions of MPM patients [36]. Therefore, an increased number of infiltrating macrophages (CD68^+^) with a M2-like phenotype (CD163^+^; CD206^+^; IL-4Rα^+^) was found in MPM and peritoneal mesothelioma [92,95]. Concurrently, TAMs communicate with MPM cells via the IL-1β/IL-1R signaling and the activation of the IL-1R pathway in tumor cells was shown to correlate with the acquisition of a CSC-like phenotype [96]. In addition, the presence of M2-like TAMs in MPM-ME was linked with an increased proliferation rate of tumor cells, such as decreased efficacy of chemotherapeutic drugs [92]. Growing evidence has shown that the release of molecules, such as IL-6, IL-10 and IL-34 by TAMs, contributed to the acquisition of a chemo/radioresistant phenotypes in tumor cells [97]. Previous studies have reported the presence of CSF-1R ligands, M-CSF and IL-34, in pleural effusion of MPM patients associated with short survival. However, only the presence of M-CSF was associated with M2-like markers expression, suggesting a different role for IL-34 in TME [98]. More recently, a chemoresistant phenotype of a CSF-1R^+^ population of mesothelioma cells, detected in primary cultures and MPM cell lines, resulted in being supported by the expression of both IL-34 and M-CSF ligands [99]. Accordingly, in vitro and in vivo studies have demonstrated that inhibition of CSF-1R might restore the CD8^+^ T cell anti-tumor response [98,100]. Inhibition of CSF-1R not only avoided mesothelioma progression and enhanced T cell response, but was also shown to increase the sensitivity of mesothelioma to PD-L1 inhibitors [101]. Additionally, in vivo studies have demonstrated the efficacy of a recently developed monoclonal antibody anti-CSF-1R (RG7155) in the reduction of CD68^+^/CD163^+^ TAMs in mesothelioma biopsies [102]. Thus, targeting the IL-34, M-CSF and CSF-1R may represent a potential therapeutical approach to suppress both mesothelioma cells and pro-tumor macrophages. TAMs may also exert immunomodulatory functions, by defending tumor cells from immune attack [103]. Lievense et al. reported the immunosuppressive properties of macrophages in MPM-ME, by a mechanism directly affecting CD4/CD8-T cell proliferation. In addition, they linked high levels of the prostanoid PGE_2_ in the pleural effusion from MPM patients with the increased number of M2-like macrophages, suggesting a potential role of PGE_2_ as promotor of a suppressive macrophage profile [61]. Miselis et al. demonstrated that targeting M2-like TAMs reduced mesothelioma growth and metastases [104]. Moreover, zoledronic acid has been reported to exhibit inhibitory functions on M2-macrophage differentiation and TAMs accumulation in mesothelioma [105]. Different strategies have been designed to reduce TAMs infiltration showing therapeutic benefits, via blocking their recruitment or direct killing, or aiming at reprogramming their anti-tumor abilities [106]. New evidence suggests that TAMs proliferate in tumors, including mesothelioma. These proliferative macrophages have been shown as a common hallmark of human solid tumors and as a potentially important prognostic marker of malignancy [107]. The mechanism that regulates TAMs self-renewal is still under investigation as a new potential therapeutic target. Little evidence exists regarding the known molecular mechanisms regulating TAMs in mesothelioma and few therapeutic strategies targeting TAMs have been developed. However, given the involvement of TAMs in drug resistance and mesothelioma progression, targeting TAMs alone or in combination with other treatments may be a promising therapeutic strategy for cancer therapy.

### 4.3. Adenosine Pathway and Mesothelioma Microenvironment

Extracellular amounts of adenosine (ADO) in tumor tissues are higher than in normal tissues because of accumulation of ATP [108]. Adenosine is an ATP-AMP metabolite that accumulates in the tumor and its expression is essentially mediated by CD39 and CD73 ectonucleotidase expression [108]. Besides tumor cells, CD39 and CD73 are also expressed on a broad range of cells of TME, such as T-cells, macrophages, MDSC, B cells, epithelial cells, and also on tumor-derived exosomes [109,110]. Extracellular ATP (eATP) is released by dying and damaged cells, and functions as immunostimulatory signal. The key event activating the adenosinergic pathway (AP) is the conversion of extracellular adenosine triphosphate (eATP) to 5′-AMP by CD39 (ectonucleoside triphosphate diphosphohydrolase-1), and then the production of ADO from 5′-AMP by CD73 (ecto-5′-nucleotidase) [111,112] (Figure 2). ADO binds and activates four different receptors (G-protein-coupled receptors A1R, A2AR, A2BR, and A3R) with different affinity and cellular specificity [111]. The AP is principally mediated by the binding of ADO to the high affinity A2AR presents on the surface of macrophages, T cells, NKs, neutrophils and dendritic cells, and on epithelial, mesothelial and cancer cells [113,114]. The balance between ATP and ADO receptors level and the expression of CD39 and CD73 ectonucleases determines the activation of an inflammatory or an anti-inflammatory response. While under physiological conditions, the AP is precisely controlled in pathologic conditions, such as cancer, the signaling is deregulated precluding autoimmunity and providing protection for malignant cells. AP in cancer, sustaining immunosuppressive cell types and thus the release of cytokines and immune modulatory factors, such as VEGF, IL6, IL10, and TGFβ, and activating survival pathways, inhibits immunosurveillance and enhances tumor survival, metastasis and therapy resistance [115]. AP deregulation in cancer has been principally linked to ADO accumulation in the TME that promotes regulatory T-cells (Tregs) activity and polarization of myeloid cells to immunosuppressive and pro-angiogenic phenotypes and affects NKs and effector T cells (CD8+), thus enhancing tumor growth. Many studies have reported that increased expression of CD73 in cancer relates to different outcomes, such as progression, poor prognosis, metastasis, and weak response to chemotherapy agents [116]. Although ADO signaling has been poorly investigated in MPM, available data support a role for AP in MPM immune-suppression. Al-Taei and collaborators [117] observed CD73 expression on TAMs in pleural effusion (PE) but not peripheral blood of mesothelioma patients or healthy donors, reinforcing the hypothesis that TAMs immunosuppressive function can involve, at least in part, ADO signaling. In addition, CD73 expression can be induced by PGE_2_, cAMP or adenosine on human CD14^+^ cells, demonstrating the existence of an autocrine loop that, upon A2AR activation, leads to the up-regulation of CD73 on human CD14+ monocytes. Activation of A2A receptors primarily has multiple inhibitory effects on the M1 macrophage subset, while adenosine receptors induce the M2 macrophage subset by up-regulating the expression of several markers, such as arginase 1 (Arg-1), tissue inhibitor of matrix metalloproteinase 1, and macrophage galactose-type C lectin 1 [118].

Several lines of evidence also support the idea that adenosine can increase VEGF secretion by macrophages through the activation of A2A receptors [119]. In addition, Cekic et al. [120] reported that A2A expression on myeloid cells, specifically TAMs, indirectly mediated suppression of T cells and NK cells in the tumor microenvironment. eATP hydrolytic activity to produce ADO has also been documented in exosomes isolated from MPM PE estimated to contribute for 20% of the total ATP-hydrolytic activity in MPM PE [121]. In addition, adenosine inhibits TNF-α and IL-12 release and augments IL-10 and vascular endothelial growth factor (VEGF) [122] production by LPS or bacteria-activated macrophages [123,124,125,126] and promotes alternative macrophage activation [127]. More recently, it has been reported that adenosine is also involved in TAMs proliferation [128]. Mechanistic studies have demonstrated that adenosine released from hepatoma cells could promote macrophages proliferation through the A2A receptor, and tumor-derived adenosine functions synergistically with autocrine GM-CSF released in TME, supporting macrophages proliferation (Figure 2). Despite the lack of information on the role of AP in MPM, these preliminary results support the rationale of investigating AP in MPM as a new therapeutic opportunity to improve the response to current therapeutic regimen and the response to the emerging immunotherapies.

**Figure 2 cancers-13-05664-f002:**
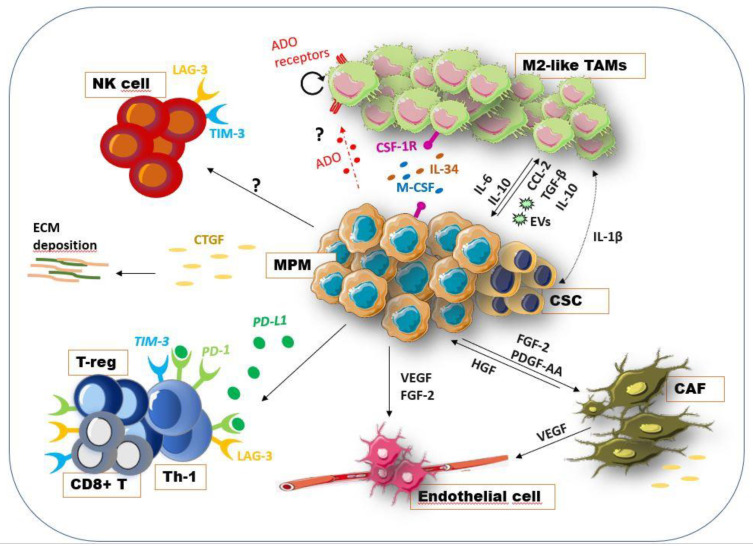
Cellular and soluble factors in MPM-ME. The interconnection between tumor cells and the surrounding stromal and immune component is necessary to create a permissive environment for cancer growth, immune escape and invasiveness [59]. Tumor cells, by the release of the growth factors FGF-2 and PDGF-AA, recruited fibroblast at tumor sites promoting their pro-tumoral activity as CAFs. In turn, these CAFs secreted the HGFs, produced ECM-related proteins, expressed the CTGF and other cytokines supporting tumor growth [65,66]. Additionally, MPM cells have been found to express the CTGF as modulator of ECM-related proteins and supporter of cancer invasiveness [62]. CAFs as well as tumor cells, via the release of the angiogenic VEGF, promoted the recruitment of endothelial cells and the vasculogenesis [67,69,70]. Exhausted Th and CD8^+^ T cells are also present in TME, characterized by the expression of immune checkpoint molecules, such as PD-1, TIM3, LAG3 [72]. Moreover, PD-L1 signaling induced Th cells reprogramming in Treg cells [73]. Moreover, the NK cells showed an immunosuppressive phenotype and low cytotoxicity [77]. TAMs, which represent the most abundant immune population [36], are recruited at tumor sites by tumor-produced CCL2 [92]. MPM cells enhance the malignancy of macrophages via the release of TGF-β, IL-10, exosomes (EVs) and M-CSF [36]. The presence of M-CSF and IL-34 was associated with short survival and chemoresistance [98,99]. Recently, the inhibition of CSF-1R has been shown to reduce mesothelioma progression and increase the susceptibility of MPM to immune checkpoint inhibitors [101]. The activation of the IL-1β/IL-1R signaling pathway in tumors by TAMs is correlated with the acquisition of a CSC-like phenotype [96]. As observed in other tumor types, adenosine (ADO) pathways may be involved in MM cells-TAMs interaction, inducing the release of pro-tumoral cytokines and promoting TAMs proliferation [115,127,128]. Although limited data indicate its involvement in MPM immunosuppression [117], a deep investigation is required. This figure was prepared using a template on the Servier medical art website (http://smart.servier.com/).

## 5. Intra-Tumor Heterogeneity within MPM Subtypes and TME

Several studies have highlighted how the immune landscape among MPM histological variants impact on the clinics and the immunotherapeutic response. Different microenvironmental stimuli drive the activation of different signaling pathways and genetic events that induce tumors to acquire distinct phenotypes and behaviors. Thus, the knowledge of the TME complexity has become important in order to understand the molecular profile of the tumor and its involvement in therapy resistance [42]. The intra-tumoral heterogeneity was investigated by Blum et al. who identified two distinct populations with epithelioid and sarcomatoid traits in different sites of MPM samples. The non-epithelioid sites resulted in being enriched in T cells, monocytes, fibroblasts, and endothelial cells as well as high expression of the immune checkpoint molecules PD-L1 and CTLA-4, determining an immunosuppressive environment [129]. Similarly, two different cellular sub-populations with distinct immunological phenotypes were discovered within MPM tumor immune microenvironment (TiME) by a CyTOF analysis on 12 tumor samples. In the study, a higher number of PD-1^+^CTLA-4^+^ CD8^+^ T cells was found in a subtype and more ICOS^+^-CTLA4^+^ T-regs and PD-1^+^ TAMs in the other one, suggesting a different response to immune-checkpoint inhibitors (ICIs) [130]. Additionally, increased macrophage infiltration was detected in the non-epithelioid MPM and a high ratio CD8^+^/CD68^+^ was correlated with a worse prognosis [34,131]. Non-epithelioid subtypes were also associated with PD-L2 positivity, and they were more likely to have a high infiltration with TIM3+ lymphocytes. Furthermore, an immune checkpoint score composition, comprehending the expression of PD-L1, PD-L2 and TIM3, when divided into three groups, identified that patients with high scores were more likely to be of non-epithelioid histology and had greater TILs [132]. On the other hand, in the epithelioid compartment, enrichment of NK cells and innate immunity markers was found, such as high prevalence of CD4^+^ and CD20^+^ cells, together with the expression of the T-cell immune suppressor VISTA [34,129,133]. As described by Yang et al., it is crucial to understand the genomic, molecular and histological heterogeneity among MPM patients given its importance at prognostic and therapeutic level and with the attempt to move to personalized treatment options [134]. Another study documented the up-regulation of VISTA on inflammatory cells in the epithelioid subtype; however, correlated it with a better overall survival. In contrast, the augmented PD-L1 expression levels in the sarcomatoid MPM were associated with a poor patient outcome [133]. Recently, Alcala et al. classified MPM types from a “hot” to “cold” tumors, based on the immune content and the overall survival [135]. Interesting data were also reported by an analysis along MPM histotypes, highlighting that T-cell immune response was progressively lost in the more aggressive sarcomatoid histotype. However, an immunosuppressive network provided by the expression of immunosuppressive cytokines and functionally impaired T-cells was also observed in the epithelioid subtype [136]. Lately, an immune-related classification of MPM patients associated with the overall survival and drug response to chemotherapeutics and ICIs was also suggested by Alay et al. [137]. Although the role of the TME network in tumor resistance mechanisms has previously reported [138], the correlation between TME heterogeneity and therapy response in MPM requires more investigation.

## 6. Therapeutic Approach Targeting TME

The standard treatment for MPM is provided by surgical resection combined with radio/chemotherapy (multimodality treatment) or radio and chemotherapy alone for unresectable tumors. However, the life expectancy of patients remains low [139]. The reduced efficacy of standard treatment is mainly related to the development of resistant mechanisms, such as the complexity and the heterogeneity of this tumor among patients [140]. Thus, a better knowledge of mesothelioma biology and chemoresistance mechanisms is required to design successful therapeutic strategies. We have previously described the known evidence about the mesothelioma microenvironment. Because most of the resistance mechanisms were mediated by the crosstalk between tumor and the surrounding microenvironmental cells via the exchange of pro-tumoral signals, cancer research has recently focused on the investigation of TME content with the aim to find new molecules to target as monotherapy or in combination with the standard treatment [36]. Different immunotherapeutic approaches for MPM have been investigated by pre-clinical and clinical studies with the aim to restore the anti-tumor immune response. Pre-clinical studies based on the use of animal models and MM cell lines have aimed to investigate mechanisms able to elicit the cytotoxicity against tumor cells and to deplete the immunosuppressive cells. These studies tested the use of antibodies targeting T regs, such as anti-CD25, or liposome encapsulated clodronate (CLIP) to reduce M2-like TAMs, and the use of dendritic cell (DC) immunotherapy or chimeric antigen receptor (CAR) T cell therapy, such as the development of chemokine/cytokine-blocking molecules [141]. The available drugs targeting TME tested in clinical trials for mesothelioma treatment are indicated in Table 1. Mesothelioma is a highly angiogenic tumor and overexpression of vascular-endothelial factor (VEGF), its receptor and other angiogenic factors, has been observed in tumor tissues. Bevacizumab is a VEGF-targeted humanized monoclonal antibody approved by FDA in different cancer types. Phase II-III clinical trials for VEGF inhibitor (Bevacizumab) used in combination with the standard chemotherapeutic drugs pemetrexed plus cisplatin showed a significant improvement in the overall survival of advanced MPM patients [142,143]. Additionally, phase II studies on the use of the VEGF, PDGF and FGF receptors inhibitor (Nintedanib) plus chemotherapeutics showed better response to therapy and improved OS, mainly for the epithelioid histologic subtypes [144]. Active immunization with DCs-conjugated with the mesothelioma antigen WT1 was tested in combination with standard chemotherapy (platinum/pemetrexed-based therapy). Phase I/II studies on DCs vaccination for MPM treatment resulted in being safe and efficacious in the stimulation of anti-cancer immune response [145]. Furthermore, DC-based immunochemotherapy was proven, with the addition of metronomic cyclophosphamide (mCTX), to deplete T reg cells and improve the anti-tumor immunomodulatory effects. The novel strategy was reported as safe and immunostimulatory, and beneficial effects in term of survival were observed in MPM patients after DC/mCTX-based treatment [146]. Mesothelioma cells are known to develop mechanisms for escaping immune surveillance, leading to T cell exhaustion via up-regulation of immune checkpoint molecules such as CTLA-4 [140]. In this regard, monoclonal antibodies targeting CTLA-4 have been developed, inducing the activation of cytotoxic T immunity. Among these inhibitors, tremelimumab and ipilimumab have been tested in clinical trials; however, no improvement in terms of overall survival for mesothelioma patients was observed [147]. Monoclonal antibodies targeting the PD-1/PD-L1 axis are also tested for mesothelioma treatment. PD-1 is generally expressed on the immune cells’ surface and once activated in response to ligand binding, they negatively regulate T cell activation. The anti-PD-1 antibodies, pembrolizumab, nivolumab and durvalumab have shown promising results in clinical trials; however, a heterogenic response among MPM patients was reported [147,148]. To increase the efficacy of these treatment modalities, combinatory strategies are under clinical investigation, with both CTLA-4 and PD-1 inhibitors as well as PD-1 inhibitors plus chemotherapy [147,149]. Other checkpoints on immune cells and up-regulated in MPM tissue as LAG-3 and TIM-3 have been recently investigated as promising targets for immunotherapy [149]. Tumor cells also increase the expression of different molecules involved in the invasive and metastatic process. Among these, TWIST1 is a key regulator of the EMT event, highly expressed by malignant mesothelioma cells. Recently, Tan et al. investigated, in pre-clinical studies, the potential of a therapeutic strategy based on the use of a DNA vaccine expressing soluble PD-1 (sPD-1) linked with TWIST1 construct, with the aim to block the immunotolerance for the self-antigen TWIST1 and destroy cancer cells. Then, because the combination of tumor vaccine and ICIs reported benefits in clinical and pre-clinical trials for cancer therapy, the hypothesis of the sPD-1-TWIST1 vaccination combined with anti-CTLA-4 was also tested, resulting in elicited long-lasting T cell immunity and mesothelioma reduction [150]. The benefits of the CAR-T cell therapy were also investigated in mesothelioma. CAR-T cells targeting the specific mesothelioma antigen surface mesothelin were evaluated, resulting in enhanced T cell activity and tumor cell destruction. However, phase I and II for CAR-T therapy are outstanding [151]. CAR-T targeting stromal factors, such as the fibroblast-activating protein (FAP) and VEGF-R2, have been developed for MPM treatment and pre-clinical data in mesothelioma mouse models showing efficacy and low toxicity [152]. Cytokine-based therapy included the intrapleural administration of interferon-α/β (IFN-α/β), resulting in partial tumor reduction and cytotoxicity, and the administration of IL-2, with therapeutic effect on pre-clinical models but controversial results, based on the route of administration, in clinical trials (intraperitoneal vs. subcutaneous/intravenous) [139,141]. The administration of emactuzumab (mAb anti-CSF-1R) was also investigated as monotherapy or combined with paclitaxel for malignant mesothelioma treatment [153]. Studies on the use of oncolytic immunotherapy demonstrated that some MPM cases, characterized by a defective IFN- α/β response, were sensitive to the oncolytic function of attenuated strains of the measles virus [154]. Additionally, mutations in the IFN-I genes correlated with CDKN2A homozygous deletion as well as oncolytic therapy response may be associated with the genetic status of BAP1. Thus, it is important to understand the defective genomic landscape in MPM patients for an appropriate therapeutic choice [155].

## 7. Conclusions

The malignant mesothelioma microenvironment is unique and the complex landscape of MPM TME is characterized by the communication mechanisms between cancer and stromal cells. Several players take part to the tumor organization and to the principal pro-tumoral functions, making tumor a complex object to analyze and understand. Different genomic defects and different immune landscape have been observed not only among patients, but also in different areas of the same tumors, highlighting the great intra-tumoral and inter-tumoral heterogeneity characterizing MPM [129]. Many interesting data correlated the malignancy of MPM TME with the infiltration of immunosuppressive TAMs [36], the high expression of immune checkpoint molecules that leads to the exhaustion of T cell activity [72], as well as the presence of CAFs promoting metastatic spread and pleural fibrosis [65,66], and the abundant content of pro-inflammatory cytokines from MPM pleural effusions [61]. Although the data reported in this review provide several information about the MPM TME players and its close connections, many questions related to their clinicopathological impact are still opened and a better knowledge of them may help cancer research for developing TME-based therapeutics able to overcome the poor overall survival of MPM patients, the chemoresistance mechanisms and the relapse of the tumor. Although the last therapy targeting TME in mesothelioma clinical trials including the immunotherapy and the combination immuno/chemotherapy are promising therapeutic approaches [148,149,150], the efficacy of these strategies remains limited. A wide consideration of TME as the main player of cancer initiation and progression is required, in order to have in-depth knowledge of TME complexity and to design new drugs that target not only tumor cells but also the TME players supporting cancer.

## Figures and Tables

**Figure 1 cancers-13-05664-f001:**
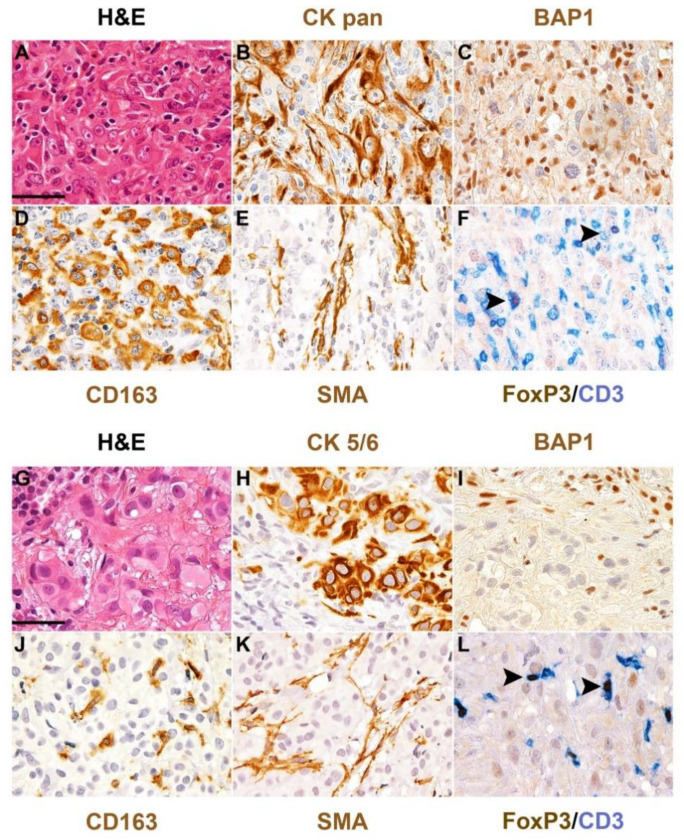
BAP1 loss and microenvironment in human mesothelioma. Hematoxylin and eosin stain and a set of immunostainings performed on formalin-fixed paraffin-embedded sections of two cases of human epithelioid mesothelioma of the pleura showing loss of BAP1. The upper case (**A**–**F**) results massively infiltrated by CD163^+^ macrophages, SMA^+^ fibroblasts and CD3^+^ T-cells including the FOXP3^+^ regulatory population, whereas the bottom case (**G**–**L**) displays a scant macrophage infiltration. Primary antibodies included broad-spectrum cytokeratin (CK, **B**) or cytokeratin 5/6 (**H**) as mesothelioma markers; anti-BAP1 (**C**,**I**); CD163 (**D**,**J**), for tumor-associated macrophages (TAMs); smooth muscle actin (SMA, **E**,**K**) for cancer-associated fibroblasts (CAFs); CD3 (**F**,**L**) for T-cells and FoxP3 for regulatory T-cells. Arrows (**F**,**L**) highlight representative regulatory T-cells. Original magnification 400×, scale bar 50 μM.

**Table 1 cancers-13-05664-t001:** Agents targeting mesothelioma microenvironment in clinical trials.

Drug Name	Therapeutic Strategy	Molecular Target	Phase	Clinical Trials	Ref
Mesothelin-targeted CAR-T	CAR-T	Mesothelin	Phase I Phase II	NCT02414269 NCT03054298	[152]
Nivolumab	mAb	PD-1	Phase I Phase II Phase III	NCT02497508 JapicCTI163247 NCT03063450	[148]
Pembrolizumab	mAb	PD-1	Phase I Phase II Phase III	NCT02054806 NCT02399371 NCT02991482	[147]
Tremelimumab	mAb	CTLA-4	Phase II	NCT01843374	[147]
Bevacizumab	mAb	VEGF		NCT00651456	[142,143]
FAP-targeted CAR-T	CAR-T	Fibroblast-activating protein (FAP)	Phase I	NCT01722149	[152]
WT1- DCV	DC vaccination	Wilms tumor protein 1 (WT1)	Phase I/II	NCT02649829	[145]
Adenoviral-mediated IFN-α/β	Cytokine-based gene therapy	Interferon	Phase I Phase I	NCT01212367 NCT01119664	[140,141]
Nintedanib	Tyrosine kinase Inhibitor	VEGF, PDGF, FGF receptors	Phase II	NCT01907100	[144]
Emactuzumab	mAb	CSF-1 receptor	Phase I	NCT01494688	[153]

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
