# Peer review of "Mesothelioma Malignancy and the Microenvironment: Molecular Mechanisms"

_cancers, 2021, doi:10.3390/cancers13225664_

Round 1

Reviewer 1 Report

In this work the authors emphasise the important role of the tumor microenvironment in malignant pleural mesothelioma for tumor progression, metastasis and therapy resistance. Every single component of the microenvironment is described in great detail and a good summary how all parts are interconnected and interact with the tumor is also given. Moreover, the authors highlight which of those parts may be potential targets for future therapy attempts.

This work may be considered for publication in Cancers after correction of  minor spelling and grammar mistakes.

Author Response

We thank you for your comments. The revised version of the manuscript with spelling and grammar checks has been uploaded. 

Reviewer 2 Report

The current manuscript aims at presenting the main components of the mesothelioma microenvironment, the most vital tumor-host cell interactions, and the perspectives of targeting these interactions for mesothelioma treatment.

The topic is not original but is relevant in the field, there are recently published two other review articles on this topic (Chu et al, 2020; Front Oncol. 2019; 9: 1366 and Hiltbruner et al; 2021, Front. Oncol., 23 June 2021 | https://doi.org/10.3389/fonc.2021.660039) but this manuscript presents treatment manipulations more extensively. No methodology needed, it is a review article. The conclusions are safely drawn and carefully made and the literature is well selected and recent.

Author Response

Thank you for your comments and suggestions. The revised version of the manuscript has been uploaded. 

Reviewer 3 Report

This work by Cersosimo, et al. summarized the interaction between MPM and its surrounding tumor microenvironment (TME) and highlighted the importance of TME as a prognostic and therapeutic potential. The work was well-structured, but some other important points related to this topic were missing. Therefore, their work can be much improved with some concerns to be addressed.

  1. As mentioned in the introduction section, MPM is characterized by a special mutational profile (BAP1, NF2, LATS1/2, CDKN2A/2B). Is there any evidence showing the TME characteristics associated with different genetic backgrounds?
  2. Also, similar to the first question. Are different histologies of MPM (epi-, bi-, sarco-) linked to distinct TME.
  3. The intra-tumoral heterogeneity of MPM also matters in terms of the TME. It would be better to include this part.

Several publications can be referred (PMID: 31945495, PMID: 32593447, PMID: 30902996, PMID: 33240401, PMID: 30322867, PMID: 31648983, PMID: 32824422, PMID: 29618661, PMID: 33632900, PMID: 32816946), but I believe there should be more. These studies also provide some potential therapeutic targets from the perspective of TME in MPM.

  1. It is nice to draw a summary graphic in Figure 2.

Many thanks for this opportunity.

Author Response

Reviewer 3: Comments and Suggestions for Authors

This work by Cersosimo, et al. summarized the interaction between MPM and its surrounding tumor microenvironment (TME) and highlighted the importance of TME as a prognostic and therapeutic potential. The work was well-structured, but some other important points related to this topic were missing. Therefore, their work can be much improved with some concerns to be addressed.

  1. As mentioned in the introduction section, MPM is characterized by a special mutational profile (BAP1, NF2, LATS1/2, CDKN2A/2B). Is there any evidence showing the TME characteristics associated with different genetic backgrounds?

Response.

Prompted by the Reviewer comment, we found in the literature that different tumor genetic background could create a mesothelioma-specific microenvironment. It has been reported that in BNC mice, where the specific disruption of the Bap1, Nf2, Cdkn2ab tumor suppressor loci in the mesothelial lining of the thoracic cavity leads to a highly aggressive MM, an infiltration of leukocytes was found [19]. In particular, a significant number of macrophages, T cells, including regulatory T cells (Tregs), B cells and NK cells was observed. That recapitulates the histological features and gene profile observed in human patients carrying combined BAP1, NF2 and CDKN2A alterations [20], indicating that the combined deletion of these tumor-suppressor genes creates a mesothelioma-specific microenvironment. The enrichment of NF-kB signaling pathway in BNC tumors likely contributes to the recruitment of immune cell to these tumors. In support of the link between TME and different genetic background, Yang H and collaborators provided evidence that CD8+ T cells were mainly enriched in MPM harboring LATS1/2 mutation compared with NF2-mutant cancer. In addition, MPM tumors harboring LATS1/2 mutation is associated with high PD-L1 expression and rather than NF2-mutant MPM display enriched Tregs signature and plasma B cell signature [21], suggesting that different tumor-infiltrating immune cell patterns exist between dysregulation of NF2 and Hippo-YAP signaling in MPM. More recently, the relationship between p14/ARF encoded by CDKN2A and tumor microenvironment was evaluated. Pezzuto et al found that p14/ARF-negative tumors are characterized by high percentage of CD163+ cells and low PD-L1 and CD4 expression, correlated with an immune microenvironment less sensitive to immune checkpoint inhibitors [22]. Collectively, all these data support the idea that genomic intratumor heterogeneity shapes the MPM tumour microenvironment and modulates host immune surveillance or immune escape in MPM [23]. This part was inserted in the Introduction section, lines 86-107. of the revised version.

  1. Also, similar to the first question. Are different histologies of MPM (epi-, bi-, sarco-) linked to distinct TME.

    3. The intra-tumoral heterogeneity of MPM also matters in terms of the TME. It would be better to include this part.

Response.

Thank you to rise these questions and we agreed that both are interesting points. In the new paragraph 5 of the revised version of our review the link between TME, intra-tumoral heterogeneity and MPM histology’s subtypes has been analysed and discussed (lines 468-501).

Several publications can be referred (PMID: 31945495, PMID: 32593447, PMID: 30902996, PMID: 33240401, PMID: 30322867, PMID: 31648983, PMID: 32824422, PMID: 29618661, PMID: 33632900, PMID: 32816946), but I believe there should be more. These studies also provide some potential therapeutic targets from the perspective of TME in MPM.

As suggested, the paragraph 6 is now updated with new references (lines 585-591).

  1. It is nice to draw a summary graphic in Figure 2.

A graphic abstract has been added.

Round 2

Reviewer 3 Report

I suggest accepting the manuscript in its current version. Congratulations to the authors for their nice work.